# Land Cover Classification in Mangrove Ecosystems Based on VHR Satellite Data and Machine Learning—An Upscaling Approach

**Neda Bihamta Toosi** [1,2,*], **Ali Reza Soffianian** [1], **Sima Fakheran** [1], **Saeied Pourmanafi** [1], **Christian Ginzler** [2] **and Lars T. Waser** [2]

1   Department of Natural Resources, Isfahan University of Technology, Isfahan 84156-83111, Iran; soffianian@cc.iut.ac.ir (A.R.S.); fakheran@cc.iut.ac.ir (S.F.); spourmanafi@cc.iut.ac.ir (S.P.)
2   Swiss Federal Institute for Forest, Snow, and Landscape Research WSL, CH-8903 Birmensdorf, Switzerland; christian.ginzler@wsl.ch (C.G.); waser@wsl.ch (L.T.W.)
*   Correspondence: n.bihamtaitoosi@na.iut.ac.ir

**Abstract:** Mangrove forests grow in the inter-tidal areas along coastlines, rivers, and tidal lands. They are highly productive ecosystems and provide numerous ecological and economic goods and services for humans. In order to develop programs for applying guided conservation and enhancing ecosystem management, accurate and regularly updated maps on their distribution, extent, and species composition are needed. Recent advances in remote sensing techniques have made it possible to gather the required information about mangrove ecosystems. Since costs are a limiting factor in generating land cover maps, the latest remote sensing techniques are advantageous. In this study, we investigated the potential of combining Sentinel-2 and Worldview-2 data to classify eight land cover classes in a mangrove ecosystem in Iran with an area of 768 km$^2$. The upscaling approach comprises (i) extraction of reflectance values from Worldview-2 images, (ii) segmentation based on spectral and spatial features, and (iii) wall-to-wall prediction of the land cover based on Sentinel-2 images. We used an upscaling approach to minimize the costs of commercial satellite images for collecting reference data and to focus on freely available satellite data for mapping land cover classes of mangrove ecosystems. The approach resulted in a 65.5% overall accuracy and a kappa coefficient of 0.63, and it produced the highest accuracies for deep water and closed mangrove canopy cover. Mapping accuracies improved with this approach, resulting in medium overall accuracy even though the user's accuracy of some classes, such as tidal zone and shallow water, was low. Conservation and sustainable management in these ecosystems can be improved in the future.

**Keywords:** ecosystem; mangrove; random forest; Sentinel-2; upscaling; Worldview-2

## 1. Introduction

Mangrove forests are considered one of the most important ecosystems on the earth. They occur in the inter-tidal zones along coasts in most tropical and semi-tropical areas [1,2]. Despite the large ecological benefits of mangrove forests, such as carbon sequestration, protection of land from erosion, purification of coastal water quality, and maintenance of ecological balance and biodiversity, mangroves have been destroyed worldwide as a result of climate change and human activities [3–6].

Qeshm Island, located off the southern coast of Iran in the Persian Gulf, is dominated by the cosmopolitan mangrove species *Avicennia marina*. Many studies have focused on the ecological and physiological characteristics of *A. marina* [7,8]. *Avicennia* species grow in oxygen-poor sediments that cannot supply the underground roots with sufficient oxygen. Consequently, their root system also includes vertically growing aerial roots (pneumatophores). These aerial roots also anchor the plants

during the frequent inundation with seawater in the soft substrate of tidal systems, and they play a significant role in sustaining mangroves [9]. Sea-level rise, a main consequence of climate change, will have a significant influence on future growing conditions [10]. Recent estimates of the extent of mangrove forests indicate that their total area has already decreased substantially, by 50% during the last half-century [11–13].

Identification of the aerial root system at a high spatial resolution would enable efficient planning of reforestation in mangrove ecosystems, but this detailed information is currently missing. Image resolution is directly correlated with the ability to identify objects of the same type [14]. Despite the great value of Landsat images for numerous applications, the specifications are inappropriate for distinguishing mudflats with aerial roots from mudflats without aerial roots. This is also due to the spectral similarities of these classes and the influence of the soil in the tidal zone (dry and wet conditions). A more detailed mapping of the mangrove ecosystem, e.g., trees and aerial root systems, is required to improve assessments of their status and recommend appropriate protection measures.

In the last years, a range of low- to high-resolution aerial images [15–17], hyperspectral images [18], Synthetic Aperture Radar (SAR) data [19], and Light Detection and Ranging (LiDAR) data [20] has been used to map the extent and distribution of mangrove cover classes. In the past decade, data have become available from Very High-Resolution (VHR) satellites, such as Worldview-2 and Pléiades-1, leading to improved mapping of mangrove cover classes [21,22]. However, the main limiting factor is the high cost of data acquisition. Consequently, alternatives have been investigated, in particular combining satellite data of different spatial resolutions [23]. Only recently, studies focusing on the use of freely available VHR data have been completed [24–28]. For example, in the forestry sector, a combination of commercially available Worldview-2 (WV-2) images and Landsat time-series data has been used to map tree species [29]. Different classification techniques, such as traditional statistical regression [30], machine learning [31], artificial neural networks [32], and tree-based methods [33,34], have successfully been used with a large geographic extent and high level of detail.

Machine learning techniques such as Random Forest (RF), artificial neural networks (ANN) and Support Vector Machine (SVM) have gained exceptional attention to classify Land cover/Land use and identify mangrove forests because they perform better than traditional techniques [33,34]. These techniques use algorithms to learn the relationship between a response and its predictors and have been categorized into two sub-types: supervised and unsupervised techniques, respectively [35]. A main advantage is that they are all nonparametric classification techniques that require no assumptions about the distribution of the data and thus no prior knowledge about the characteristics of feature data is needed either [31]. Many studies in the field of Land cover/Land use classification have been carried out using different machine learning algorithms as well as comparing them among each other [35]. In the last decade, RF has recently become preferred for mapping land cover classes in several realms [36,37]. RF is a nonparametric technique based on a set of decision trees. Unlike parametric techniques, RF can be used to predict land cover classes even based on a small sample size and therefore reduces both cost and time [38]. Moreover, embedded feature selection in the model generation process makes it possible to obtain high mapping accuracy. Several studies have demonstrated that RF, in combination with satellite data (Landsat) [37] and a high spatial resolution [16], can be used to successfully map mangrove cover classes. Moreover, the latest advances in remote sensing data and techniques, i.e., increasing availability of datasets in combination with higher temporal, spatial and spectral resolutions (e.g., ESA Copernicus Program Sentinel-1/-2), enable improved characterization of mangrove ecosystems. They make it possible to derive leaf area index, height and biomass, map the mangrove forest extent, and monitor mangrove status over time [39]. Several studies have been carried out to explore satellite data of different spatial resolutions for improving land cover maps, i.e., in forestry that have combined data sets from Landsat and AVHRR [40] or Landsat and MODIS [41]. However, to the best of our knowledge, no study exist that combine Worldview-2 and Sentinel-2 images to classify mangrove ecosystems in greater detail which is a prerequisite for managing this ecosystem. Therefore, freely available Sentinel-2 data, in combination with commercially available

high-spatial-resolution imagery, has great potential for mapping wall-to-wall mangrove cover at a high level of detail, i.e., distinguishing between land cover classes with similar spectral properties.

In the present study, we investigated whether the combination of Sentinel-2 and Worldview-2 imagery can be used to accurately map the most relevant land cover classes for mangrove ecosystem management. We developed a three-step approach: (i) extraction of reflectance values from high-resolution Worldview-2 imagery, (ii) segmentation based on spectral and spatial features, and (iii) wall-to-wall mapping of the eight land cover classes based on Sentinel-2 imagery.

The study aims at developing a cost-effective, accurate method that can be applied widely and in a standardized manner, particularly when field surveys are restricted.

## 2. Materials and Methods

In order to produce a wall-to-wall map of mangrove cover classes for Qeshm Island, a two-step method was applied: (i) Reference data were generated at a 0.5-m spatial resolution using an object-based method performed on Worldview-2 images. The Worldview-2 data were dispersed across the entire study area and covered 27% of the total land cover. (ii) Reference data based on Worldview-2 images were used for the upscaling.

### 2.1. Study Area

Qeshm Island is located a few kilometers off the southern coast of Iran, opposite the port cities of Bandar Abbas and Bandar Khamir. It is the largest island in the Persian Gulf and covers an area of 1491 km² (Figure 1). Most of the mangrove forests of Qeshm are located in the northern part of the island in the Hara Protected Area, a biosphere reserve that covers an area of approximately 20 by 20 km and is characterized by numerous tidal channels [42]. The mangroves are rooted in the saltwater of the Persian Gulf, but the special pores within their leaves extract the salt from the water. The whole forest area is affected by frequent boat trips, fishing and a small amount of leaf-cutting for livestock feed. The forests are the habitat for migratory birds, hooked turtles and venomous aquatic snakes, all of which are indigenous species.

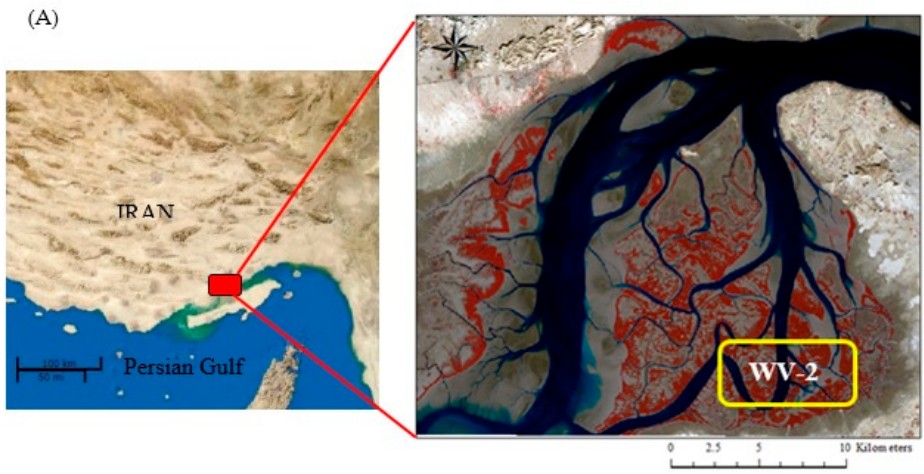

**Figure 1.** *Cont.*

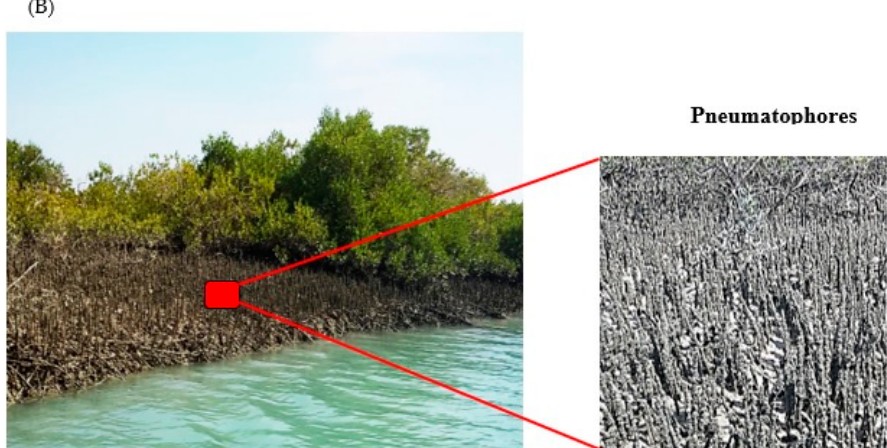

**Figure 1.** (**A**) Left: location of Qeshm Island and the mangrove ecosystem, shown as a false-color Sentinel-2B image (2017, Combination of Bands 8-4-3); right: Worldview-2 image data used for the upscaling approach. (**B**) Aerial roots (pneumatophores) growing in a wide radius around the mangrove (*Avicenna marina*) are highlighted by the red polygon.

### 2.2. Field Data

The field survey revealed that *Avicenna marina* was the dominant mangrove species on Qeshm Island. Visual analysis of high-resolution images made it possible to distinguish between eight target classes of mangrove ecosystem, including three types of mangrove spatial pattern: closed canopy mangrove, open canopy mangrove, and individual mangrove trees (found in a small patch on the island). The remaining target classes in the study area were mudflat (either with or without aerial roots), tidal zone (sand, beaches or unvegetated area), shallow water (rivers or ponds), and deep (open) water.

During the field survey, a total of 170 GPS reference points (Garmin 629sc with spatial accuracy between 1 and 5 m) were collected and used for validation of the classification of the eight land cover classes. In order to minimize and avoid the negative impacts on the vulnerable ecosystem, the collection of field samples was restricted to easily accessible parts. In order to increase the number of samples for three types of mangrove and two types of mudflat, 53 points were additionally selected from Spot 6/7 data using image interpretation. Figure 2 shows the distribution of the samples for the eight land cover classes. The set of reference points collected from both GPS and from the Spot images are depicted for each class separately in Table 1.

**Table 1.** Overview of the two different sets of reference points collected from the GPS survey and the Spot 6/7 image interpretation.

| Source of Reference Points | Land Cover Class | | | | | | | | |
|---|---|---|---|---|---|---|---|---|---|
| | Closed Mangrove Cover | Open Mangrove Cover | Individual Mangrove Trees | Mudflats | Aerial Roots | Tidal Zone | Shallow Water | Deep Water | Total |
| GPS | 5 | 12 | 0 | 27 | 7 | 6 | 7 | 28 | 92 |
| Spot 6/7 images | 12 | 15 | 11 | 15 | 25 | 0 | 0 | 0 | 78 |
| Total | 17 | 27 | 11 | 42 | 32 | 6 | 7 | 28 | 170 |

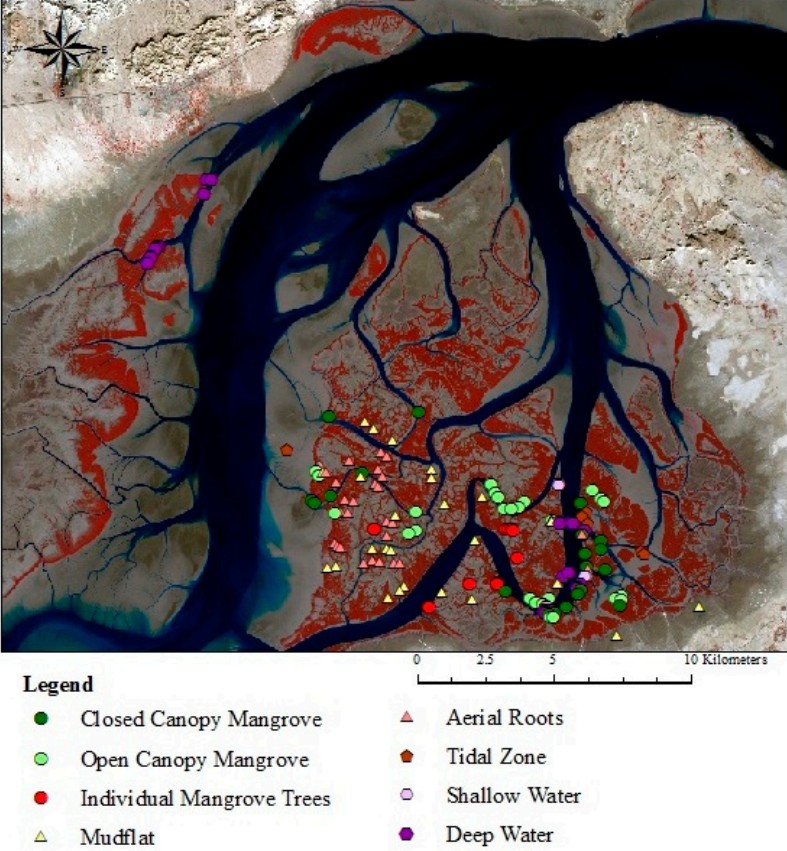

**Figure 2.** Distribution of the samples obtained from the field survey and from Spot 6/7 image interpretation of the whole study area.

## 2.3. Remote Sensing Data and Pre-Processing

Technical specifications of the Worldview-2 and Sentinel-2 imagery are given in Table 2. Images were cloud-free over coastal areas. The multispectral bands of Worldview-2 consist of four standard bands (red, green, blue and near-infrared 1) and four additional bands (coastal, yellow, red edge and near-infrared 2), which facilitated spatial and spectral analysis, mapping and monitoring of large areas at a more detailed level [43]. Sentinel-2 bands consist of four bands at a 10-m spatial resolution (blue, green, red and near-infrared), six bands at a 20-m spatial resolution (four narrow bands near the red edge and two wider SWIR), and three bands at a 60-m spatial resolution (aerosols, water vapor and cirrus) [44]. The obtained data were pre-georeferenced to the UTM zone 40 North projection using the WGS-84 datum. Sentinel-2 data were radiometrically calibrated to apparent surface reflectance by the FLAASH (Fast Line-of-sight Atmospheric Analysis of Hypercubes) atmospheric corrected algorithm [45] in ENVI 5.4 software. Fusion of panchromatic with multispectral images of Worldview-2 data resulted in an image with a 0.5-m spatial resolution. In the present study, the Gram Schmidt pan-sharpening algorithm was applied [46] because it preserves the primary spectral value of the objects and has successfully been applied to multispectral images. In this study, a Sentinel-2 level 1C product image was applied, acquired on a clear day and under the lowest tide condition over Qeshm Island.

## 2.4. Spectral Variability

VHR images show the required details of the mangrove ecosystem. Therefore, the Worldview-2 image was used to select the eight targeted land cover classes: (1) closed canopy mangrove, (2) open canopy mangrove, (3) individual mangrove trees, (4) mudflats, (5) aerial roots, (6) tidal zone, (7) shallow water, and (8) deep water. In order to better separate them and distinguish between the spectral

signatures, based on the field survey and image interpretation, reflectance values of the target classes (by 100 points) were extracted from Worldview-2 image bands. The boxplots in Figure 3 show that the two classes closed canopy mangrove and open canopy mangrove are clearly distinguished by the blue band and the yellow band. Moreover, it shows that the aerial roots are clearly distinguished from the mudflats in the green, yellow and red bands. Figure 4 shows the reflectance values of the eight land cover classes for the Sentinel-2 bands.

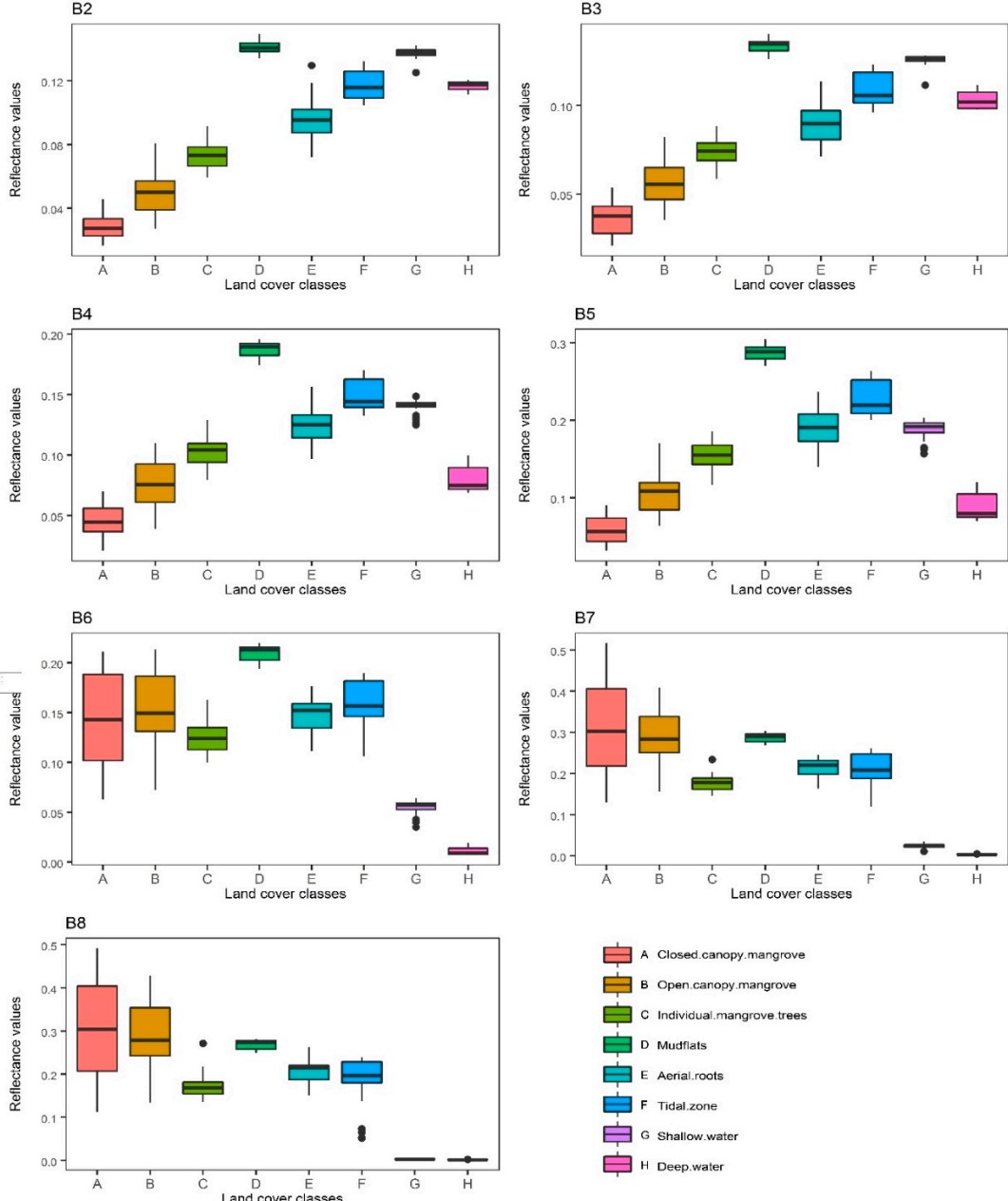

**Figure 3.** Reflectance values of the eight land cover classes for each Worldview-2 band: (**B2**) (Blue: 450–510 nm), (**B3**) (Green: 510–580 nm), (**B4**) (Yellow: 585–625 nm), (**B5**) (Red: 630–690 nm), (**B6**) (Red edge: 705–745 nm), (**B7**) (Near-infrared 1: 770–895 nm), and (**B8**) (Near-infrared 2: 860–1040 nm). The letters A to H show the land cover classes namely closed canopy mangrove, open canopy mangrove class, individual mangrove trees, mudflats, aerial roots, tidal zone, shallow water, and deep water, respectively.

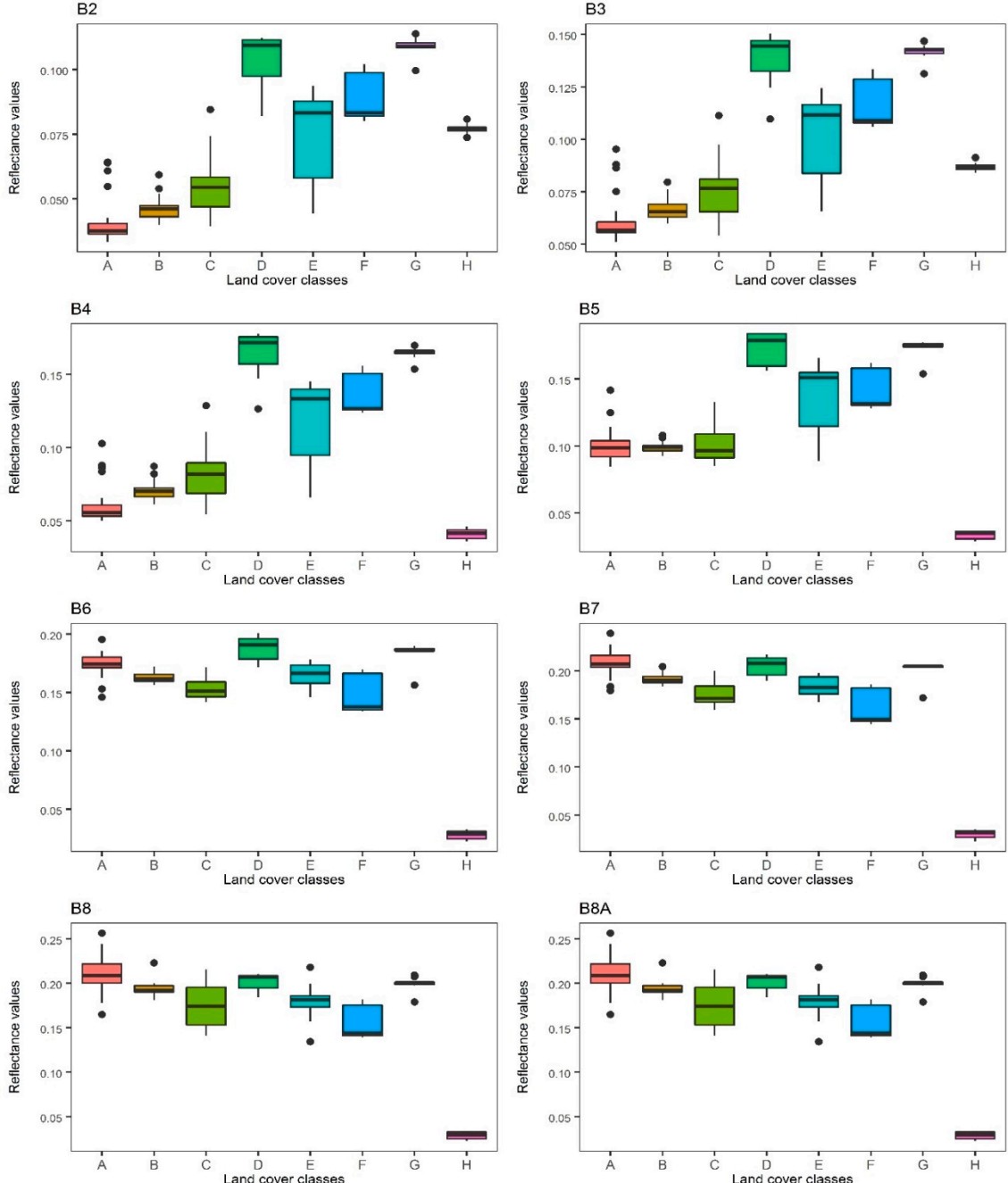

**Figure 4.** Reflectance values of the eight land cover classes for Sentinel-2: (**B2**) (Blue band 490 nm), (**B3**) (Green band 560 nm), (**B4**) (Red band 665 nm), (**B5**) (Vegetation Red Edge band 705 nm), (**B6**) (Vegetation Red Edge band 740 nm), (**B7**) (Vegetation Red Edge band 783 nm), (**B8**) (Near-infrared band 842 nm), and (**B8A**) (Vegetation Red Edge band 865 nm). The letters A to H show the land cover classes namely closed canopy mangrove, open canopy mangrove class, individual mangrove trees, mudflats, aerial roots, tidal zone, shallow water, and deep water, respectively.

**Table 2.** Sensor specifications of the Worldview-2 and Sentinel-2 imagery.

| Sensor | Worldview-2 | Sentinel-2 |
|---|---|---|
| Acquisition date | 26.12.2016 | 02.12.2017 |
| Bands | 8 multispectral<br>1 panchromatic | 13 multispectral |
| Spatial resolution | 2 m<br>0.5 m | 10 m (bands: 2, 3, 4, 8)<br>20 m (bands: 5, 6, 7, 8A, 11, 12)<br>60 m (bands: 1, 9, 10) |
| Dynamic range | 11 bits | 12 bits |
| Swath width | 16.4 km at nadir | 290 km |
| Revisit time | 1.1 day | 10 days |

### 2.5. Reference Data

The sampling of reference data used Object-Based Image Analysis (OBIA), which is based on segmentation [34,47]. The multi-resolution segmentation algorithm from eCognition 9.2 software (Trimble Inc., Munich, Germany) [48] was used, which classifies homogeneous image objects by using attributes of image objects rather than the attributes of individual pixels or a hierarchical object-oriented approach using a knowledge base. In the present study, a series of scale parameters, shape and compactness (from low to high) were tested to control the size of segmentation. In order to generate reliable reference samples, information from the Normalized Difference Vegetation Index (NDVI) layer and the Moran Index using the Worldview-2 bands was additionally included for image segmentation. In previous studies, NDVI has been successfully applied to display and quantify mangrove forest changes [12,49,50]. NDVI values were computed as:

$$\text{NDVI} = \frac{\text{NIR} - \text{Red}}{\text{NIR} + \text{Red}} \tag{1}$$

where NIR is band 8 and Red is band 5.

The Moran index provides the correlation between attributes at each location in a study area and the statistical mean of the values from neighboring locations. The Moran index has successfully been applied in almost all studies dealing with spatial autocorrelation (for a review see [51]). It evaluates the magnitude of homogeneity of a target image object to other objects surrounding it. If targets are attracted to (or repelled from) each other, the observations are dependent [52]. In addition, the Moran Index is similar to correlation coefficients and its value ranges from −1 to 1 [53]. Moreover, the Moran index provides quantitative clustering information that is used to select homogeneous regions. The Moran index measures the degree of spatial auto-correlation at each particular location [54]. Information and photos from the field observations, as well as a visual interpretation of Worldview-2 images, were used to develop the rule sets to select segmentations for each class as reference data (Ground Truth or OBIA training). In order to use spectral features (mean and standard deviation of blue, yellow, red edge bands and NDVI), additional geometric features such as shape and extent were used. The total number of variables selected was based on visual inspection of the reflectance values of the eight classes. The feature selection process was completed with the eCognition feature optimization tool using 100-point datasets.

### 2.6. Upscaling by Reference Data

After the generation of the reference data, RF was used to classify Worldview-2 and Sentinel-2 images. In this step, Sentinel-2 imagery was preliminarily mapped over the same extent as the Worldview-2 image with 70% of the reference data. The accuracy of the map of the RF algorithm was then checked, and the reference data were used for mapping mangrove classes to a larger extent. A layer stack was created from the NDVI, blue, green, red and near-infrared bands. Sentinel-2 data (10 m spatial resolution) served as input for the RF classification. RF was performed using the *Ranger*

Package in the R statistical software [55]. Figure 5 shows the main steps of the classification approach applied in this study.

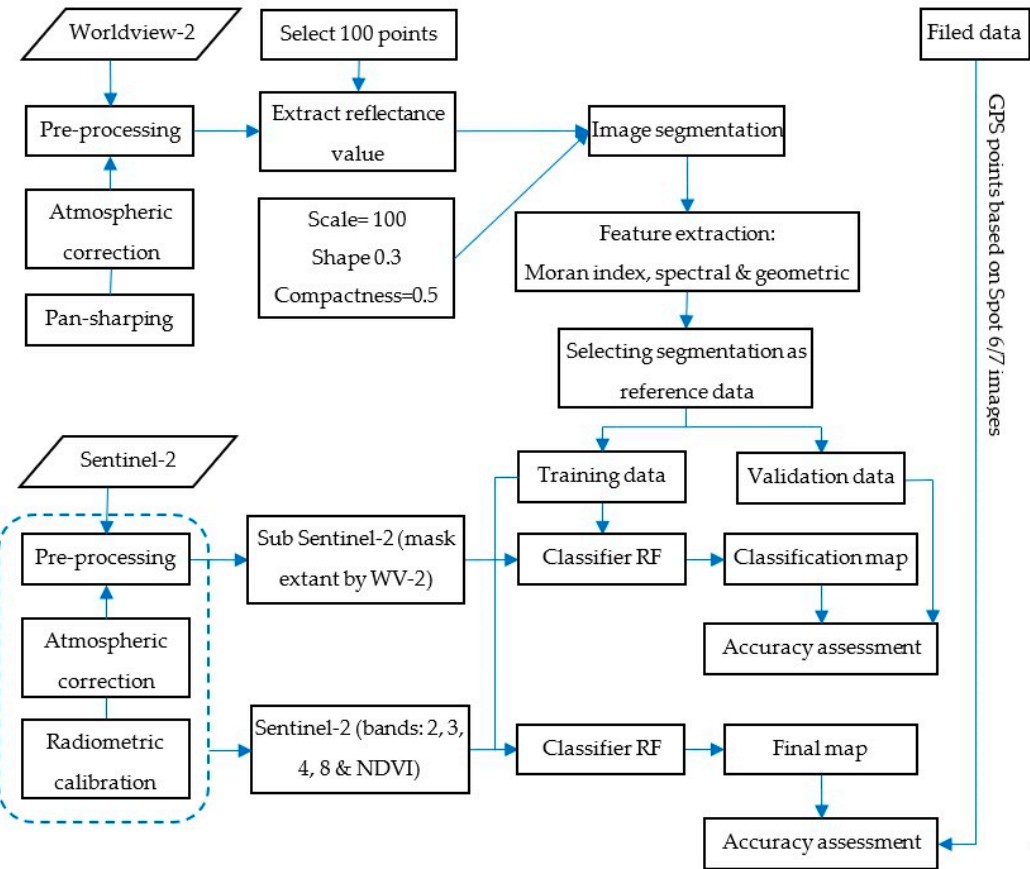

**Figure 5.** Flow chart of the upscaling approach for mapping land cover in mangrove ecosystems.

### 2.7. Accuracy Assessment

The land cover map based on the classification using Sentinel-2 images (same extent as Worldview-2) was assessed using 30% of the reference data, which was excluded from classification and from cross-validation. To assess the accuracy of the land cover map based on the upscaling approach, a confusion matrix was constructed, consisting of 167 validation points collected during the field survey, for image interpretation using the Spot 6/7 images. We used a leave-one-out cross-validation [56] because our sample was relatively small (30% of the training data did not cover the land cover map to a large extent).

This matrix provides the overall accuracy, the kappa coefficient, and the user's and producer's accuracies for each class. The producer's accuracy represents how well reference pixels of the ground cover type are classified. The validation points were rasterized to the 10-m resolution of the Sentinel-2 image. Furthermore, a Wilcoxon test (non-parametric statistical test that compares two paired groups) was applied in order to estimate the significance difference between the user's and producer's accuracies for the two classification maps [57].

### 3. Results

The mapped land cover classes of the Qeshm Island mangrove ecosystem are given in Figure 6.

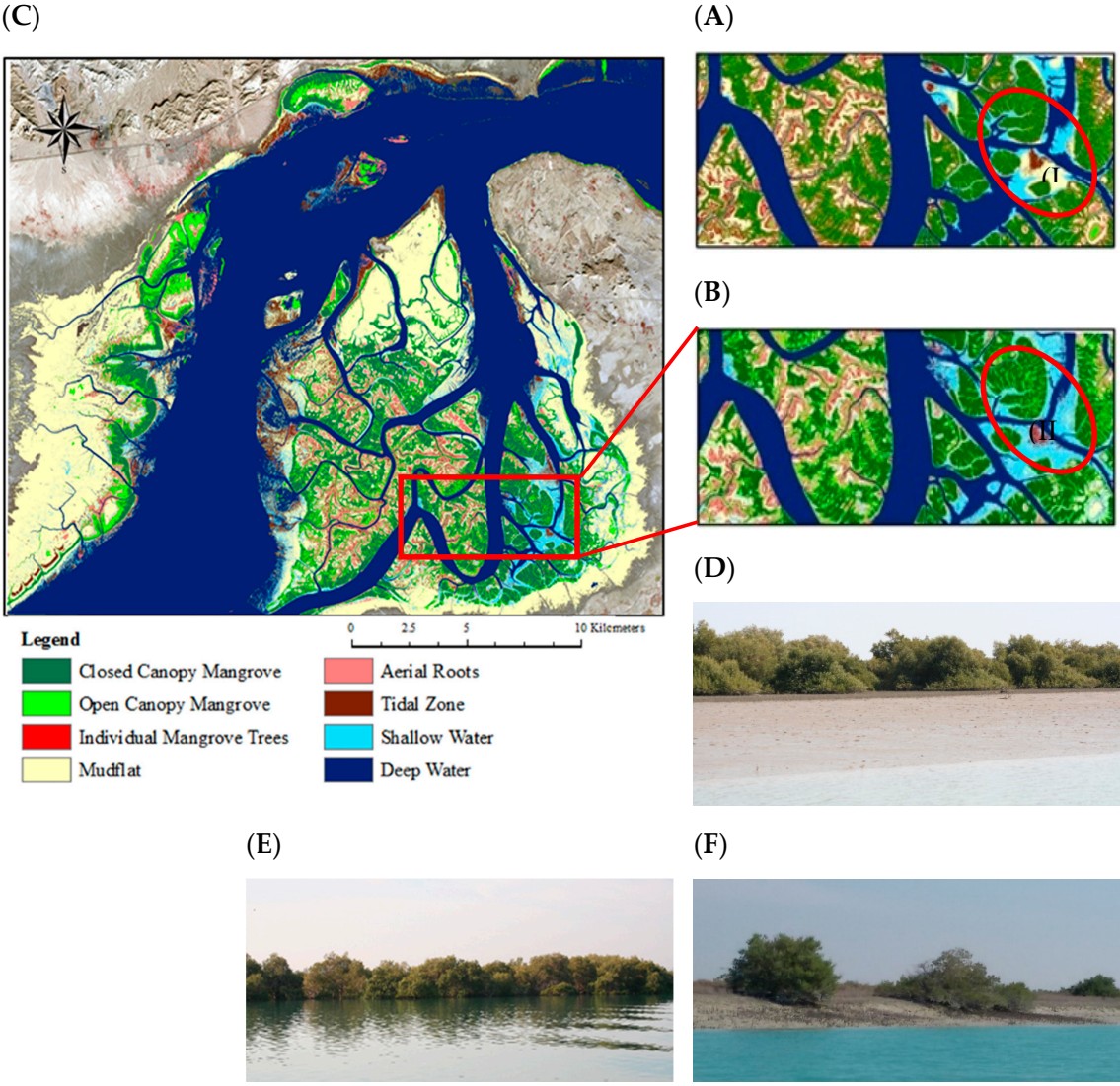

**Figure 6.** Classification map of the Worldview-2 image (**A**), classification map of the Sentinel-2 image with the same extent as the Worldview-2 image (**B**), and the final map based on the upscaling approach (**C**). The visible differences between (I) and (II) are related to misclassified shallow water. This error happened two reasons: First, the spectral profiles of the shallow water and tidal zone classes were similar in the Sentinel-2 image. Second, the date of the images differed, and the relative sea level rise had acted as an important factor in converting the tidal zone class (**D**) to shallow water (**E**). However, we were able to show more details of mangrove ecosystems with this approach, such as individual trees and aerial roots (**F**).

Model accuracies of the RF classification were assessed in two steps. In the first step, random reference data based on the segmentation of Worldview-2 images was used to validate the subset of Sentinel-2 imagery. An overall accuracy of 88% and a kappa coefficient of 0.85 were obtained. The validation revealed the producer's accuracy of the four classes shallow water (96.5%), deep water (94.8%), closed canopy mangrove (89.2%), and mudflat (83.1%) (Figure 7). In the second step of the validation, the overall accuracy of the upscaling approach was calculated at 65.5% and the kappa coefficient was 0.63. Whereas the user's accuracy for the two classes deep water (100%) and closed canopy mangrove (75.1%) was high, the producer's accuracy for the class mudflat with aerial roots (66.1%) and without aerial roots (73.3%) were lower (Figure 8). These two classes included a corollary omission error of 33.9% and 26.7%, respectively. The results of the confusion matrix of the different classification extents are given in Tables 3 and 4.

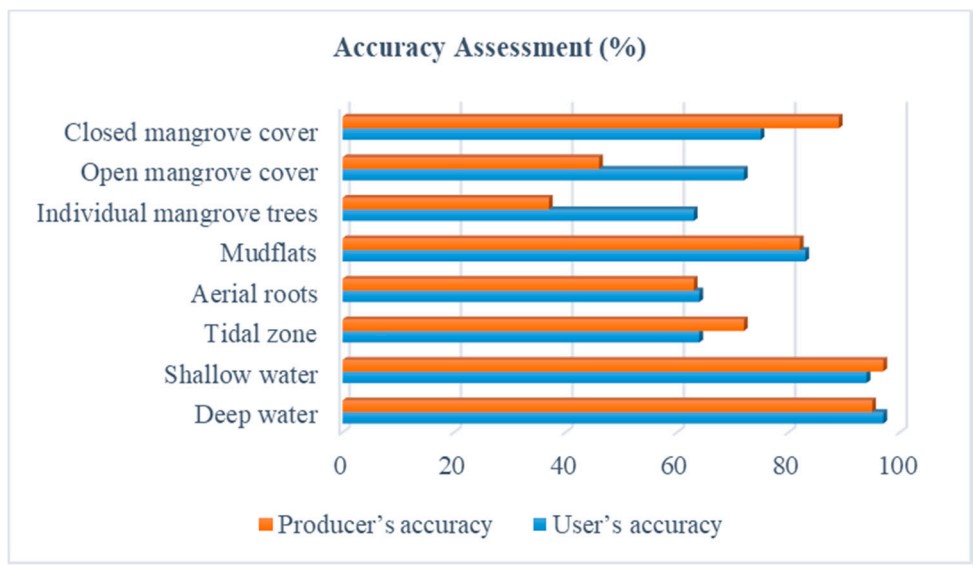

**Figure 7.** Accuracy statistics of the classification map of Sentinel-2 over the same extent as for Worldview-2.

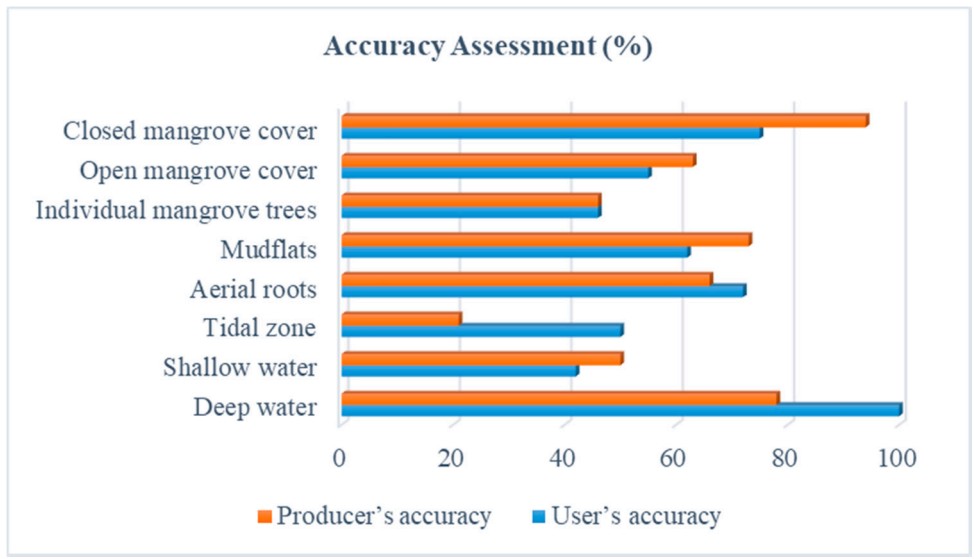

**Figure 8.** Accuracy statistics of the classification map of the upscaling approach.

**Table 3.** Confusion matrix for the classification map of Sentinel-2 over the same extent as for Worldview-2. Bold-faced numbers indicate the agreement between a class.

| Classification | Reference Data | | | | | | | |
|---|---|---|---|---|---|---|---|---|
| | Closed Mangrove Cover | Open Mangrove Cover | Individual Mangrove Trees | Mudflats | Aerial Roots | Tidal Zone | Shallow Water | Deep Water |
| Closed mangrove cover | **1102** | 119 | 235 | 1 | 3 | 0 | 0 | 2 |
| Open mangrove cover | **73** | **696** | 122 | 4 | 61 | 5 | 1 | 1 |
| Individual mangrove trees | 61 | 59 | **23** | 1 | 6 | 4 | 1 | 2 |
| Mudflats | 0 | 4 | 0 | **991** | 90 | 113 | 2 | 0 |
| Aerial roots | 0 | 160 | 19 | 38 | **682** | 163 | 2 | 0 |
| Tidal zone | 0 | 28 | 3 | 158 | 232 | **738** | 3 | 0 |
| Shallow water | 0 | 6 | 0 | 4 | 7 | 2 | **1049** | 51 |
| Deep water | 0 | 0 | 0 | 0 | 0 | 0 | 28 | **1042** |

**Table 4.** Confusion matrix for classification map of the upscaling approach. Bold-faced numbers indicate the agreement between a class.

| Classification | Reference Data | | | | | | | |
|---|---|---|---|---|---|---|---|---|
| | Closed Mangrove Cover | Open Mangrove Cover | Individual Mangrove Trees | Mudflats | Aerial Roots | Tidal Zone | Shallow Water | Deep Water |
| Closed mangrove cover | **15** | 2 | 0 | 0 | 0 | 0 | 0 | 0 |
| Open mangrove cover | 0 | **15** | 1 | 2 | 1 | 5 | 3 | 0 |
| Individual mangrove trees | 1 | 1 | **5** | 3 | 1 | 0 | 0 | 0 |
| Mudflats | 0 | 1 | 5 | **22** | 10 | 3 | 0 | 1 |
| Aerial roots | 0 | 2 | 0 | 2 | **23** | 4 | 0 | 1 |
| Tidal zone | 0 | 0 | 0 | 1 | 0 | **3** | 0 | 2 |
| Shallow water | 0 | 0 | 0 | 0 | 0 | 0 | **3** | 4 |
| Deep water | 0 | 0 | 0 | 0 | 0 | 0 | 0 | **28** |

The classification revealed that the largest area (27,678 ha) belongs to the class deep water and smallest (62 ha) to the class individual mangrove trees. The classes closed canopy mangrove, open canopy mangrove, mudflat, aerial root and tidal zone cover an area of 4857, 3474, 13,099, 2296, and 2026 ha, respectively. The *p*-value of the Wilcoxon test for differences in the user's and producer's accuracies between the two classification maps were 0.11 and 0.32, respectively, which is greater than the significance level alpha = 0.05. We can conclude that the accuracy assessments did not differ significantly between the two classification maps.

## 4. Discussion

### 4.1. General Comments

Mangrove forests typically grow in zones that are marshy and inaccessible [11]. Therefore, collecting GPS points as training data through field surveys is difficult [14]. Nowadays, new developments in remote sensing techniques have great potential to overcome the problem of acquiring field data in inaccessible areas of mangrove ecosystems [58]. Between 1970 and 2018, approximately 435 studies mapping the area of mangroves were conducted, and after the year 2000 the majority used Landsat images [14]. While Landsat imagery has the advantages of free availability, a large archive and extensive coverage, its relatively coarse spatial resolution of 30 m can be a major limitation. The potential of different datasets from Landsat, ALOS AVNIR-2, Worldview-2 and LIDAR to map a detailed land cover of mangrove ecosystems was recently evaluated [59]. The results clearly demonstrated the importance of a higher spatial resolution for mapping specific mangrove features, such as individual tree crowns and species communities.

With the present study, we contribute to this research with an efficient mapping of mangrove features using multi-resolution datasets. We add to existing knowledge gained in a previous study [37], which focused on comparing four classification algorithms based on Landsat images for predicting six land cover classes in the mangrove ecosystem: mangrove forest, mud flat, other land cover, tidal zone, water and settlement. The results of this earlier research demonstrated that using Landsat data enables to potentially distinguish between different mangrove forest stands and can be useful for detecting their changes over time. However, since mangrove forests usually consist of small patches, Landsat images are not suitable for extracting more details and are mainly only appropriate for detecting changes in mangrove forest canopies. This is in accordance with [14,59], in that only high-resolution images can be used to map more detailed land cover classes. By increasing the number of spectral bands and the spatial resolution, it is possible to discriminate between small objects and to detect small objects, such as individual trees and mudflats with aerial roots. Several studies have shown the potential of Worldview-2 data for detailed land cover mapping, including mangrove forest ecosystems [16,39,59–61]. However, the main reason for the limited use of such imagery is its high

cost–in particular, for developing countries. Thus, in our study, an upscaling approach was applied that reduces costs while still enabling the generation of a more detailed map of land cover classes.

### 4.2. Modelling Approach

In the last decade, several studies have been carried out combining satellite data of different spatial resolutions to improve land cover maps in the forestry sector. Some investigations have considered the combination of Landsat data with datasets of higher spatial resolutions such as IKONOS [62], GeoEye-1 data [63] or Worldview-2 [29].

Comparison of the two confusion matrices clearly demonstrated that the accuracy and kappa of the upscale approach were lower than the accuracy and kappa of the map that had the same extent as the one based on the Worldview-2 imagery. The use of a large amount of reference data to predict the subset of Sentinel-2 data helped to reduce misclassification.

The confusion matrix of the upscaling approach (Table 4, Figure 8) indicates that the overall accuracy and kappa decreased with increasing map scale. There was a high incidence of misclassification of individual trees and tidal zone when Sentinel-2 data were used. Several possible reasons for this error exist. First, it might be due to the amount of reference data because the Worldview-2 data only cover about 27% of the Sentinel-2 image. On the other hand, in the Worldview-2 image, the area of these two classes is less than that of the other classes. It is well known that the number of reference samples from the Worldview-2 image affects classification accuracy. In a recent study, it was demonstrated that the large amount of reference data obtained from the Worldview-2 image was the main driving factor for the accuracy of the classification of two pine tree species by Landsat data [29]. Future work could include the collection of more training samples in order to further improve the distinction of these land cover classes. Second, the error could be a result of the similarity of the spectral profile of individual trees and open canopy mangrove forest. The use of fewer reference samples decreases the spectral separability of classes and potentially decreases the accuracy. Third, the decrease in accuracy could be related to the level pre-processing and viewing geometry of Sentinel-2 imagery.

Nevertheless, the present study demonstrates that areas with different canopy densities and mudflat areas (occurrence of aerial root systems) can be accurately classified using the upscaling approach with Sentinel-2 images. Overall, high accuracies were obtained for mapping closed canopy mangrove (75% user's accuracy, 94% producer's accuracy) and aerial roots (72%, 66%). Moreover, the combined use of Worldview-2 and Sentinel-2 images further increases map accuracies–in particular when the overall accuracy is not very high, and the user's accuracy is low in problematic classes.

### 4.3. Importance of Mapping of Detailed Information on Mangrove Forests

Detailed maps of mangrove ecosystems are a prerequisite for successful protection and management. Since mangroves occur in areas with a high salt concentration in the soil, they have developed aerial roots for physiological functions and cover a large area within the Hara Protected Area [64]. This specialized root system reduces the power of sea waves and guarantees sustainable establishment of mangrove communities, as well as providing a protected place for aquatic animals [42]. In order to plan the development of mangrove forests, both naturally or artificially, the selection of potential suitable land is relevant. The land areas on the map that show the mangrove forests and mudflat with aerial roots are preferred to other areas that are not covered by vegetation. Moreover, the occurrence of mangrove is an indication that the land provides optimal conditions for the development of mangrove forests in terms of soil parameters such as salinity and pH. Mapping the details of mangrove ecosystems is an effective way to visualize, evaluate and better understand mangrove ecosystem development. Changes over a long period, as well as the recognition of unexpected changes due to natural or dramatic anthropogenic impacts, can be assessed at an early stage [65,66]. Moreover, assessing changes in the aerial root area can indicate the status of these forests because these roots are destroyed by an increase in water level or sediments.

## 5. Conclusions

In the present study, we demonstrate that field surveys in mangrove ecosystems are not always feasible, due to the high costs and inaccessibility of the area. Mangrove distribution mapping is a hot topic in the field of mangrove remote sensing [14]. Based on field observations, the mangrove forests in the present study have a uniform composition of the species *Avicenna marina* and the detectable differences are limited to canopy density, which consists of mangrove zonation patterns including forests of immature trees and of mature trees, and isolated trees. The use of VHR satellite imagery for sampling reference data in combination with freely available satellite data and machine learning is an effective and straightforward approach to further improve the details of land cover maps and to assess relevant forest parameters. Upscaling is a cost-efficient tool for producing accurate large-scale land cover maps in inaccessible ecosystems. The findings of the present study support the sustainable management of mangrove ecosystems and can be used to assess the efficiency of ecosystem services. Although the upscaling approach produced low user accuracies for the shallow water and tidal zone classes, overall accuracies were generally high.

With the proposed method, it is possible to distinguish between the two most relevant classes for management, i.e., canopy mangrove canopy and mudflat. Our findings confirm that advances in remote sensing data and techniques are favorable for developing novel methods to map mangrove ecosystems in greater detail. We conclude that the selection of appropriate images remains an important factor and that Sentinel-2 images have great potential for identifying different land cover types, thanks to their high spatial, temporal and spectral resolution. Continuity of the presented approach is guaranteed since Sentinel-2 data will be continuously acquired.

**Author Contributions:** N.B.T. is responsible for the study. N.B.T. developed the methodology, programming, statistical analysis and was the main writer of the manuscript. L.T.W. supported the development of the approach and methodology and writing the manuscript. C.G. supported the development of the approach and methodology. A.R.S., S.F. and S.P. conceived and provided useful suggestions to the manuscript. All authors have read and agreed to the published version of the manuscript.

**Funding:** This research received no external funding.

**Acknowledgments:** This study was carried out in the framework of the doctoral thesis and was supported by the Department of Natural Resources, Isfahan University of Technology, Isfahan, Iran. We thank Melissa Dawes for professional language editing.

**Conflicts of Interest:** The authors declare no conflict of interest.

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
