# Peer review of "Land Cover Classification in Mangrove Ecosystems Based on VHR Satellite Data and Machine Learning—An Upscaling Approach"

_remotesensing, doi:10.3390/rs12172684_

Round 1

Reviewer 1 Report

General comments

This paper contains very interesting data, in particular, those where the information contained is of particular international relevance. The paper investigated the potential of combining Sentinel-2 and Worldview-2 data to classify eight land cover classes in a mangrove ecosystem in Iran. This new approach intends to minimize the costs of acquiring commercial satellite data for collecting reference data and to focus on freely available satellite data for mapping land cover classes of mangrove ecosystems. The approach and findings of the paper sound to be an effective and straightforward approach to further improve the details of land cover maps and to assess relevant forest parameters.

This thematic is presently considered a high concern topic by scientists, policy makers, governmental bodies, non-governmental organizations and the general public.

  1. The text of paper shows clarity and brevity.
  2. The manuscript included recent literature reviews to assess the current state of knowledge regarding new developments in remote sensing techniques
  3. New statements and state of the art is included.
  4. Data treatment and analysis were
  5. Graphics and tables format and layout are adequate
  6. This paper research provides a scientific contribution of great potential for identifying different land cover types

Author Response

Responses to comments of reviewer 1

We thank you for taking the time to review our manuscript, we are pleased that you liked our study and we sincerely appreciate your statement and valuable comments.

Comments to authors:

General comments

       This paper contains very interesting data, in particular, those where the information contained is of particular international relevance. The paper investigated the potential of combining Sentinel-2 and Worldview-2 data to classify eight land cover classes in a mangrove ecosystem in Iran. This new approach intends to minimize the costs of acquiring commercial satellite data for collecting reference data and to focus on freely available satellite data for mapping land cover classes of mangrove ecosystems. The approach and findings of the paper sound to be an effective and straightforward approach to further improve the details of land cover maps and to assess relevant forest parameters.

     This thematic is presently considered a high concern topic by scientists, policy makers, governmental bodies, non-governmental organizations and the general public.

The text of paper shows clarity and brevity.

The manuscript included recent literature reviews to assess the current state of knowledge regarding new developments in remote sensing techniques.

New statements and state of the art is included.

Data treatment and analysis were.

Graphics and tables format and layout are adequate.

This paper research provides a scientific contribution of great potential for identifying different land cover types.

Response:

Thank you for your comments.

Reviewer 2 Report

This paper proposed an upscaling approach to classify 8 land covers in the mangrove ecosystem. Authors use 170 reference points collected from the VHR data to validate their work.

Some comments are summarized below:

  1. Most land cover classification methods for remote sensing are based on pixel. An image data will have thousands of pixels. Then the classification model will use some portion of pixels to train and the rest to test. So is 170 reference points enough to validate the concept in this study?
  2. Although random forest has been used for classification in some studies, there are many other classification models such as support vector machine, k-nearest neighbourhood, etc. Why not testing their performance in your study?
  3. Is there any state-of-the-art methods for mapping land cover in mangrove ecosystem? Could you make a comprehensive comparison?
  4. The proposed method has a complicated workflow, it sometimes is hard to follow. It is better to illustrate some key stage for better presentation.
  5. The main classification stage seems follow a standard routine, i.e. using NDVI and some bands as spectral feature, followed by a classifier. Therefore, the novelty of this study is not clear.
  6. The overall accuracy is just 65.5%, which seems to be insufficient.
  7. Figure5 is illustrated poorly, please improve it.
  8. There is too much space left in page 12 and 13, please restructure the paper and make it tidy.
  9. The font of this paper is not consistent, please check page 3

Author Response

Responses to comments of reviewer 2

We thank reviewer 2 for taking the time to review our manuscript. We sincerely appreciate your fruitful and valuable comments, which helped to improve the manuscript substantially. Below you will find our point-by-point answers which are highlighted in blue in the manuscript.

Comments to authors:

This paper proposed an upscaling approach to classify 8 land covers in the mangrove ecosystem. Authors use 170 reference points collected from the VHR data to validate their work.

Some comments are summarized below:

  1. Most land cover classification methods for remote sensing are based on pixel. An image data will have thousands of pixels. Then the classification model will use some portion of pixels to train and the rest to test. So is 170 reference points enough to validate the concept in this study?

Response:

  • Thank you for these comments. We have clarified the text. First, we focused on a subset of the Sentinel-2 image for the prediction which resulted in more than 2000 points (30% reference data from WV-2) to perform the accuracy assessment. Then, we used 170 independent reference points for the accuracy assessment. These points didn’t cover the subset of the sentinel-2 image.
  • Moreover, more extended fieldwork is not always feasible because of the inaccessibility of the protected and flooded areas within the mangrove ecosystem. Nevertheless, we added a suggestion that future work could include more reference (field samples) in order to further improve the distinction between the land cover classes.

  1. Although random forest has been used for classification in some studies, there are many other classification models such as support vector machine, k-nearest neighborhood, etc. Why not testing their performance in your study?

Response:

  • Principally, we know that e.g. according to Jensen (2009), the success of a classification rather depends on appropriate training data than on the classification algorithm chosen. Based on review studies (see e.g. Fassnacht et al. 2016, RSE) we believe that RF and SVM are favorable for land cover classification compared to other machine-learning techniques (Ma,2017; Mountrakis, 2011,). In a previous study we examined different classification algorithms and RF performed best in model and prediction accuracies classifiers for mangrove mapping (Toosi, 2019). The results of One-way ANOVA clearly revealed no significant difference between classifiers for mapping mangrove classes and RF performed better than other machine learning algorithms. Although SVM is often used for classification, it did not perform as well as RF and is often very slowly regarding computation time (see e.g. Waser et al. 2017, Remote Sensing).

  1. Is there any state-of-the-art methods for mapping land cover in mangrove ecosystem? Could you make a comprehensive comparison?

Response:  

  • Two methods for mapping land cover in the mangrove ecosystem are generally used. The first is solely based on using ground data whereas the second uses remote sensing data. Traditionally, large-scale mangrove mapping was limited to sketch maps, fieldwork maps, and the digitizing of analogue datasets (Spalding, 1997) which are disadvantageous in terms of cost and time.
  • Recently, the number of studies using remotely sensed data to assess extent, change, ecosystem structure, ecosystem services, and vulnerability of mangroves have been established (Kuenzer, 2011). The first global map exclusive to mangrove forests that is based on the single usage of remotely sensed data was from Giri et al. (2011). While using remote sensing data are more feasible in terms of time, cost, and integrated visibility than solely using field surveying data, the latter is more accurate. Only recently has the availability of VHR satellite increased the accuracy of using remote sensing data.

  1. The proposed method has a complicated workflow, it sometimes is hard to follow. It is better to illustrate some key stage for better presentation.

Response: 

  • We have adapted figure 5 accordingly and believe that the workflow of the up-scaling approach is now better to follow.

  1. The main classification stage seems follow a standard routine, i.e. using NDVI and some bands as spectral feature, followed by a classifier. Therefore, the novelty of this study is not clear.

Response: 

  • We agree with the arguments that vegetation indices such as NDVI and in RS spectral information is frequently used in classifying land cover. However, the novelty of the proposed method is to minimizing the cost of accessing commercial satellite data for reference data collection (which is not feasible in inaccessible ecosystems such as the mangroves without destroying them) and focusing on freely available satellite data (Sentinel-2). Moreover, reproducibility and up-dating of the land cover maps are guaranteed due to the huge archive of Sentinel-2 data. Future work will include Deep learning approaches and focus on the entire use of non-commercial software. We also believe, and this has already been confirmed by feedback form authorities, that the presented approach has a high practical relevance.

  1. The overall accuracy is just 65.5%, which seems to be insufficient.

Response:

  • Thank you for your comment, we do not entirely agree that 65% are overall insufficient. Several studies include “easy to classify” classes to increase overall accuracies - which we didn’t do. In the Discussion section "modelling approach" we explained in detail how to interpret overall accuracy (lines 337-354). The decrease in overall accuracy for the final map is due to the classes tidal zone that have a very small area in the Worldview-2 image and therefore the amount of reference data is not enough to process this classes in the Sentinel-2 image. We know that this is a drawback of the presented approach. Nevertheless, the present study demonstrates that areas with different canopy densities and mudflat areas (occurrence of aerial root systems) can be accurately classified using the upscaling approach with Sentinel-2 images. This information and findings are relevant for a sustainable management and protection of the mangroves. Overall, high accuracies were obtained for mapping closed canopy mangrove (75% user’s accuracy, 94% producer’s accuracy) and aerial roots (72%, 66%) - two very relevant classes in the mangrove ecosystem.

  1. Figure5 is illustrated poorly, please improve it.

Response:

  • Thank you for your comments, we have the improved figure 5 accordingly (page 10).

  1. There is too much space left in page 12 and 13, please restructure the paper and make it tidy.

Response:

  • Thank you for your comments. We have removed spaces on pages 12 and 13 by restructuring the text.

  1. The font of this paper is not consistent, please check page 3

Response:

  • Thank you for your comments. We have harmonized the font size throughout the entire manuscript.

References:

1- Ma, L.; Li, M.; Ma, X.; Cheng, L.; Du, P.; Liu, Y. A review of supervised object-based land-cover image classification. ISPRS J. Photogramm. Remote Sens. 2017, 130, 277–293.

2-. Mountrakis, G.; Im, J.; Ogole, C. Support vector machines in remote sensing: A review. ISPRS J. Photogramm. Remote Sens. 2011, 66, 247–259.

3- Toosi, N.B.; Soffianian, A.R.; Fakheran, S.; Pourmanafi, S.; Ginzler, C.; Waser, L.T. Comparing different classification algorithms for monitoring mangrove cover changes in southern Iran. Glob. Ecol. Conserv. 2019, doi:10.1016/j.gecco.2019.e00662.

4- Spalding, M., F. Blasco, C. Field. World Mangrove Atlas. International Society for Mangrove Ecosystems, WCMC, National Council for Scientific Research, Paris, 1997.

5- Fassnacht, F.E.; Latifi, H.; StereĹ„czak, K.; Modzelewska, A.; Lefsky, M.; Waser, L.T.; Straub, C.; Ghosh, A., 2016: Review of studies on tree species classification from remotely sensed data. Remote Sensing of Environment, 186: 64-87.

6- Waser, L.T.; Ginzler, C.; Rehush, N., 2017: Wall-to-wall tree type mapping from countrywide airborne remote sensing surveys. Remote Sensing, 9, 8: 766 (24 pp.).

7- Kuenzer, C., A. Bluemel, S. Gebhardt, T. V. Quoc, S. Dech. Remote sensing of mangrove ecosystems: A review. Remote Sensing, 2011, 3, 878-928.

8- Giri, C., et al. Status and distribution of mangrove forests of the world using earth observation satellite data. Global Ecology and Biogeography, 2011, 20(1), 154-159.

Reviewer 3 Report

The article presents an interesting application of land use classification utilizing an upscaling approach and random forest methods.

I include below some comments for improving the presentation of the article.

-------------------------------------------------------------------------

Specific comments.

Abstract.

 km2 please use superscript for "2" (squared).

line 25-25 "Since cost is a limiting factor in generating land cover maps, the latest remote sensing techniques are advantageous." This sentence should appear earlier in the introduction, not after results.

line 29-31 should appear before results.

Introduction

Paragraphs 42-64 have a very local scope, and few relevant research background information for a global audience. In contrast, the background in the use of machine learning, and random forest in particular, for land cover classification, is very briefly discussed in lines 78-89. I would strongly suggest to shorten the first more local section (42-64), and instead, expand the background in the use of machine learning, and random forest in particular, for land cover classification in general, and also for mangrove in particular.

One of the interesting aspects of the article is the use of upscaling from Worldview to Sentinel. Background in the use of upscaling in land cover literature should be summarized in the introduction.

The introduction should stress the knowledge gaps in the literature. Why is this article an important contribution to the literature of machine learning for land cover prediction?

Methods.

study area. It might be convenient to add some references to the study area if available.

line 130. Please mention the spatial accuracy of the GPS positioner utilized.

Figure 3. Perhaps add letters A-H to the simbology showing land cover classes in the bottom right for greater clarity.

Figure 5. Please correct text so that it fits the diagram shapes.

Where is the use of Moran Index included in figure 5?

Discussion.

Discussion section "general comments" probably makes a better job in introducting the need for upscaling studies from worlview to sentinel than the introduction. I would suggest to move some of these comments to the introduction section to better justify the need for the current study earlier in the article.

Similarly, lines 313-317 should be better included in the introduction.

Conclusions.

Discussion and conclusions could include needs for future work.

Author Response

Responses to comments of reviewer 3

We thank reviewer 3 for taking the time to review our manuscript. We sincerely appreciate your fruitful and valuable comments, which helped to improve the manuscript substantially. Below you will find our point-by-point answers which are highlighted in red in the manuscript.

Comments to authors:

Abstract.

  • km2 please use superscript for "2" (squared).
  • line 25-25 "Since cost is a limiting factor in generating land cover maps, the latest remote sensing techniques are advantageous." This sentence should appear earlier in the introduction, not after results.
  • line 29-31 should appear before results.

Response:

  • Km2 is changed to km2 (line=23).
  • This sentence has been moved to lines 19-21.
  • These sentences have been moved to lines 25-27.

Introduction

  • Paragraphs 42-64 have a very local scope, and few relevant research background information for a global audience. In contrast, the background in the use of machine learning, and random forest in particular, for land cover classification, is very briefly discussed in lines 78-89. I would strongly suggest to shorten the first more local section (42-64), and instead, expand the background in the use of machine learning, and random forest in particular, for land cover classification in general, and also for mangrove in particular.
  • One of the interesting aspects of the article is the use of upscaling from Worldview to Sentinel. Background in the use of upscaling in land cover literature should be summarized in the introduction.
  • The introduction should stress the knowledge gaps in the literature. Why is this article an important contribution to the literature of machine learning for land cover prediction?

Responses:

Thank you for your comments.

  • We have removed this information from lines 42-64 and instead added the background in the use of machine learning for land cover classification in lines 79-89. The required background information for RF is now given in lines 92-95.
  • We have summarized required background information for using upscaling and added it to lines 99-103.
  • We have added more information about the need of novel RS technologies in endangered ecosystems and underscored the potential of machine learning in land cover classification studies. See lines 83-92 and 103-106, respectively.

Methods.

  • study area. It might be convenient to add some references to the study area if available.
  • line 130. Please mention the spatial accuracy of the GPS positioner utilized.
  • Figure 3. Perhaps add letters A-H to the symbology showing land cover classes in the bottom right for greater clarity.
  • Figure 5. Please correct text so that it fits the diagram shapes.

Where is the use of Moran Index included in figure 5?

Responses:

Thank you for your comments.

  • We have added a new reference (Number reference:42- Zahed et al. 2010).
  • The spatial accuracy of the GPS position is between 1 and 5 meter (subpixel) and now given in the text (lines= 144-145).
  • We have changed figure 3 accordingly.
  • We have changed the figure 5 accordingly and have added the Moran Index as well.

Discussion.

  • Discussion section "general comments" probably makes a better job in introducing the need for upscaling studies from worldview to sentinel than the introduction. I would suggest to move some of these comments to the introduction section to better justify the need for the current study earlier in the article.

Similarly, lines 313-317 should be better included in the introduction.

Responses:

Thank you for your comments.

       11- We have added background information about the utility of upscaling in the introduction section while underscoring general comments on the importance of this approach. On other hand, if this section is added to the introduction, the introduction section will be too long and we think that the background in the use of upscaling can help to better justify the need for the current study earlier in the article.

-The suggested lines 313-317 have been modified and moved to lines 101-103.

Conclusions.

      12- Discussion and conclusions could include needs for future work.

Response:

Thank you for your comments. We have added some needs for future work in the discussion (Lines 344-345) and in the conclusion (Lines 391-392).

Round 2

Reviewer 2 Report

The authors have addressed all my comments. I am satisfied with the revised manuscript.